# SUPERVISED DIMENSION CONTRASTIVE LEARNING

## ABSTRACT

Self-supervised learning has emerged as an effective pre-training strategy for representation learning using large-scale unlabeled data. However, models pre-trained with self-supervised learning still require supervised fine-tuning to achieve optimal task-specific performance. Due to the lack of label utilization, it is difficult to accurately distinguish between positive and hard negative samples. Supervised contrastive learning methods address the limitation by leveraging labels, but they focus on global representations, leading to limited feature diversity and high cross-correlation between representation dimensions. To address these challenges, we propose Supervised Dimension Contrastive Learning, a novel approach that combines supervision with dimension-wise contrastive learning. Inspired by redundancy reduction techniques like Barlow Twins, this approach reduces cross-correlation between embedding dimensions while enhancing class discriminability. The aggregate function combines the embedding dimensions to generate predicted class variables, which are optimized to correlate with their corresponding class labels. Orthogonal regularization is applied to ensure the full utilization of all dimensions by enforcing full-rankness in the aggregate function. We evaluate our method on both in-domain supervised classification tasks and out-of-domain transfer learning tasks, demonstrating its superior performance compared to traditional supervised learning, supervised contrastive learning, and self-supervised learning methods. Our results show that the proposed method effectively reduces inter-dimensional correlation and enhances class discriminability, proving its generalizability across various downstream tasks.

## 1 INTRODUCTION

Recent advances in self-supervised learning (Chen et al., 2020; Zbontar et al., 2021; Caron et al., 2020; 2021) have demonstrated its effectiveness as a pre-training method using large-scale unlabeled data. This approach leverages data augmentation to generate semantically similar examples and then aligns the representations of the two examples. Despite its success, models pre-trained through self-supervised learning still require fine-tuning with labeled data to achieve sufficient performance on specific tasks. Moreover, relying solely on data augmentation and sample discrimination limits the ability to distinguish between positive and hard negative samples. effectively (Robinson et al., 2020; Wu et al., 2020).

With the growing availability of large labeled datasets, e.g. JFT-300M (Sun et al., 2017), supervised representation learning has gained importance. By directly utilizing labels, supervised methods can capture semantically meaningful relationships between data points without heavily depending on data augmentation. Contrastive learning has shown promising results in supervised frameworks, similar to its success in the field of self-supervised learning. Supervised contrastive learning methods (Khosla et al., 2020; Zha et al., 2024; Cui et al., 2021; 2023) encourage representations of data points from the same class to cluster closely while pushing apart representations from different classes in the representation space. It demonstrated superior performance in classification tasks compared to both self-supervised learning and traditional cross-entropy-based approaches.

Despite these advantages, supervised contrastive learning methods primarily focus on global representations and do not induce the learning of additional discriminative features, once different classes are sufficiently separated. This results in a lack of feature diversity, as evidenced by the high cross-correlation between dimensions of the learned representations in Figure 1. Consequently, supervised contrastive learning methods may underperform in tasks that require diverse feature representations

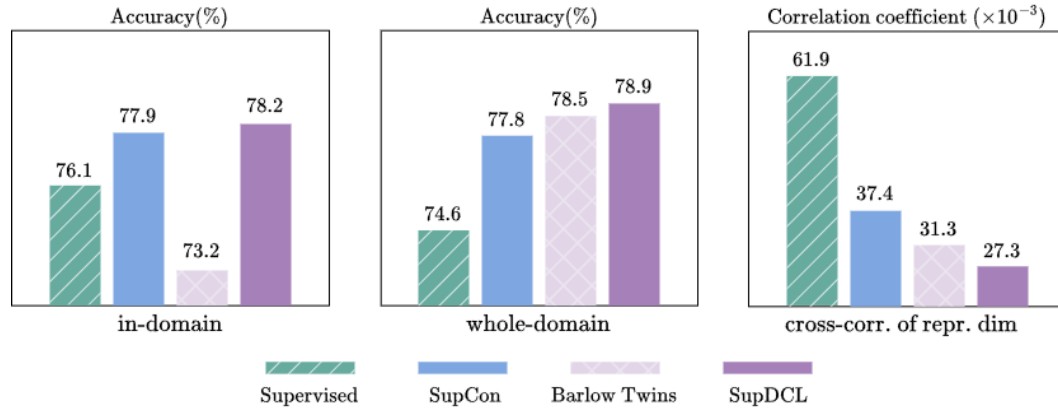

Figure 1: **Evaluating representation diversity and generalization across learning methods.** From left to right, we present the in-domain classification Top-1 accuracy on ImageNet-1K, the average Top-1 accuracy across in-domain and 10 out-domain downstream datasets, and the mean of off-diagonal cross-correlation values between dimensions in the representation space. These results are based on a ResNet50 backbone pre-trained on ImageNet-1K using various methods.

such as transfer learning. To address these limitations, supervised representation learning must capture a wide range of discriminative features that are semantically meaningful. These features should maximize class separability while maintaining diversity within the learned representations.

To achieve diverse feature learning, we leverage Barlow Twins (Zbontar et al., 2021), a redundancy reduction-based self-supervised learning method. Barlow Twins operates by decorrelating embedding dimensions to learn more diverse features across different embedding dimensions. We describe such approaches as dimension contrastive learning.

In this work, we propose a method called Supervised Dimension Contrastive Learning (SupDCL), a novel way of applying supervision to dimension contrastive learning. By incorporating supervision, SupDCL reduces cross-correlation between embedding dimensions while simultaneously enhancing class discriminability within each dimension. Specifically, we introduce a new concept called *discriminativeness*, which reflects how strongly each embedding dimension contributes to class separability. Our method ensures that each dimension is trained to be discriminative, meaning it correlates with specific class variables, enhancing its ability to distinguish between different classes. We employ the aggregate function combining the embedding dimensions to generate predicted class variables that are trained to correlate with their corresponding class labels. To ensure that the aggregate function utilizes all dimensions without exclusion, we apply orthogonal regularization. It enforces the function to be full-rank thereby leveraging the entire embedding space.

We evaluate our method on both in-domain supervised classification tasks, where the dataset and task are defined by the same labels used during training, and out-domain transfer learning tasks, where the dataset and task differ from those encountered during pre-training. Our results show that SupDCL outperforms self-supervised learning methods in in-domain classification tasks and achieves results comparable to supervised contrastive methods. Moreover, in out-domain transfer learning, SupDCL consistently outperforms all other methods, demonstrating its superior generalization capability across various downstream tasks. The main contributions of this work are summarized as follows:

- We identify a limitation in existing supervised representation learning, specifically in supervised contrastive learning methods, which suffers from high cross-correlation between dimensions, leading to limited feature diversity and suboptimal generalization.

- We introduce SupDCL, a novel method that applies supervision to dimension contrastive learning, reducing inter-dimensional correlation while enhancing discriminative ability.

- The proposed method outperforms self-supervised learning methods for in-domain classification tasks and achieves comparable results to supervised contrastive methods. For out-domain transfer learning, SupDCL surpasses all methods, demonstrating its ability to generalize effectively across diverse datasets.

## 2 RELATED WORKS

### 2.1 SELF-SUPERVISED LEARNING

Recent advancements in self-supervised learning leverage the potential of unlabeled data, drawing significant attention to various methodologies (Chen et al., 2020; He et al., 2020; Caron et al., 2020; Grill et al., 2020; Chen & He, 2021; Caron et al., 2021; Zbontar et al., 2021; Bardes et al., 2021). Self-supervised methods are typically categorized into three main directions: contrastive, non-contrastive, and dimension contrastive learning, each addressing robust learning challenges.

Contrastive learning focuses on instance-level pairwise similarity by attracting positive pairs (augmented versions of the same instance) and repelling negative pairs (different instances). Methods like SimCLR (Chen et al., 2020) show the effectiveness of such learning but require large batches to provide sufficient negatives. MoCo (He et al., 2020) addresses this issue with a memory bank and momentum encoder for efficient negative sampling. These approaches excel at learning representations invariant to augmentations, enabling strong out-domain transfer performance.

Non-contrastive methods (Grill et al., 2020; Chen & He, 2021; Caron et al., 2021) avoid negatives, addressing the collapse problem where representations converge to trivial solutions. BYOL (Grill et al., 2020) and SimSiam (Chen & He, 2021) use asymmetric architectures with predictors and stop-gradient mechanisms to prevent collapse. DINO (Caron et al., 2021) aligns student-teacher networks via knowledge distillation, emphasizing consistency without contrastive objectives.

Dimension contrastive learning methods (Zbontar et al., 2021; Bardes et al., 2021; Caron et al., 2021) focus on relationships between embedding dimensions rather than instances. Barlow Twins (Zbontar et al., 2021) minimizes correlation between dimensions to enforce independence, while VICReg (Bardes et al., 2021) combines invariance, variance, and covariance terms to balance dimension-level and global relationships. The duality between contrastive and dimension contrastive learning (Garrido et al., 2022) highlights their complementary strengths.

### 2.2 SUPERVISED LEARNING

Supervised learning methods commonly rely on cross-entropy loss for classification tasks. However, its limitations, including sensitivity to class imbalance and reliance on instance-specific predictions, are well-documented (Elsayed et al., 2018; Cao et al., 2019; Zhang & Sabuncu, 2018). Studies propose alternatives involving label distribution modifications (Müller et al., 2019; Szegedy et al., 2016) and advanced augmentations like MixUp (Zhang et al., 2017).

Contrastive-based supervised learning extends self-supervised methods by incorporating class information. SupCon (Khosla et al., 2020) generalizes contrastive learning by treating all same-class samples as positive pairs, improving generalization compared to cross-entropy. PaCo (Cui et al., 2021) and GPaCo (Cui et al., 2023) propose class-wise learnable centers, validating their effectiveness under class-balanced and imbalanced settings. Despite these advancements, supervised representation learning approaches often fail to address feature redundancy and diversity in learned representations, limiting their out-domain performance.

Our work bridges this gap by proposing a novel framework that incorporates dimension decorrelation and class-specific discriminativeness to learn generalizable features. Unlike SupCon, which emphasizes global relationships between instances within the same class, our method optimizes dimension-level independence while explicitly maximizing class-related information, achieving strong transfer performance comparable to self-supervised methods.

## 3 METHOD

We generate two augmented views of each image in the input batch $X$ using augmentations $t^A, t^B \sim \mathcal{T}$, resulting in $X^A$ and $X^B$. These augmented views are then passed through the encoder network $f$ to obtain representations $Y^A$ and $Y^B$ respectively. Instead of directly applying the training loss to these representations, we first pass them through the projector network $g$ to produce embeddings $Z^A = (F_1^A, \ldots, F_M^A)$ and $Z^B = (F_1^B, \ldots, F_M^B)$, each with an embedding dimension of $M$. The training loss is applied to these embeddings, as is common in self-supervised learning methods.

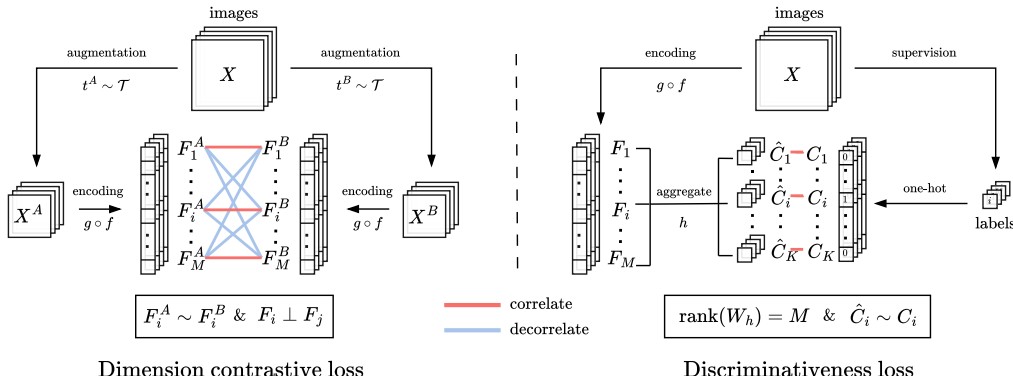

Figure 2: **The overall framework of Supervised Dimension Contrastive Learning (SupDCL).** The left side illustrates dimension contrastive learning, which reduces redundancy through decorrelation and enforces augmentation invariance across dimensions. The right side shows the discriminativeness learning process, where an aggregate function predicts class variables that are correlated with their true class labels. The aggregate function is constrained to be full-rank, ensuring that all embedding dimensions contribute to the prediction. By combining these processes, SupDCL enhances both redundancy reduction and discriminative power in the learned representations.

## 3.1 SELF-SUPERVISED DIMENSION CONTRASTIVE LOSS

According to previous study (Garrido et al., 2022), dimension contrastive learning methods such as Barlow Twins (Zbontar et al., 2021) and VICReg (Bardes et al., 2021) can be categorized under a unified objective. Our approach builds upon the formulation of Barlow Twins, a representative method in dimension contrastive learning. The left side of Figure 2 illustrates the objective of Barlow Twins, which is also defined as:

$$\mathcal{L}_{\text{DCL}} \triangleq \lambda_{\text{inv}} \underbrace{\sum_{i=1}^{M} (1 - \mathcal{C}_{ii})^2}_{\text{invariance}} + \lambda_{\text{decorr}} \underbrace{\sum_{i=1}^{M} \sum_{j \neq i}^{M} \mathcal{C}_{ij}^2}_{\text{decorrelation}} = \lambda_{\text{inv}} \mathcal{L}_{\text{inv}} + \lambda_{\text{decorr}} \mathcal{L}_{\text{decorr}}. \tag{1}$$

Here, the cross-correlation matrix $\mathcal{C}$ is computed between two embeddings, $Z^A$ and $Z^B$, along the batch dimension. By treating each dimension of the embeddings as a random variable, the embeddings themselves can be viewed as vectors composed of these variables. The cross-correlation between embedding dimensions is then represented as:

$$\mathcal{C}_{ij} = \text{Cor}[F_i^A, F_j^B] = \frac{\text{Cov}[F_i^A, F_j^B]}{\sqrt{\text{Var}[F_i^A]\text{Var}[F_j^B]}} \quad \text{for } i, j \in [M], \tag{2}$$

where the covariance and variance of embedding dimensions are computed empirically using the samples within the batch. The invariance loss ensures that the cross-correlation between corresponding dimensions of embeddings from different views converges to one, promoting augmentation invariance within each embedding dimension. In contrast, the decorrelation loss minimizes the correlation between different embedding dimensions, aiming to reduce redundancy and enhance feature diversity across dimensions.

## 3.2 SUPERVISED DIMENSION CONTRASTIVE LOSS

### 3.2.1 CLASS CORRELATION

For supervised pre-training, it is crucial to ensure that the embedding dimensions learned from self-supervised dimension contrastive loss are both augmentation-invariant and decorrelated, while also being discriminative for classification. To achieve this, we encode class labels into a one-hot vector of size $K$, where $K$ represents the number of classes. This allows us to treat each dimension $C_i \in \{0, 1\}$ as a *class variable* representing the corresponding class $i$.

We define an *aggregate function* $h : \mathbb{R}^M \to \mathbb{R}^K$, which takes the embedding $Z = (F_1, \ldots, F_M)$ as input and produces the predicted class variables $\hat{C} = (\hat{C}_1, \ldots, \hat{C}_K)$ as output. To measure the alignment between predicted class variables $\hat{C}_i$ and their corresponding true class variables $C_i$, we calculate the correlation between each pair of predicted and true class variables. We refer to this as the *class correlation* $d_i$, which is formulated as:

$$d_i = \text{Cor}[C_i, \hat{C}_i] = \frac{\text{Cov}[C_i, \hat{C}_i]}{\sqrt{\text{Var}[C_i]\text{Var}[\hat{C}_i]}} \quad \text{for } i \in [K]. \tag{3}$$

By maximizing the class correlation, we encourage the embedding dimensions to be not only augmentation-invariant and decorrelated but also discriminative for the supervised classification task. To formalize this alignment, we introduce the *class correlation loss*, which is defined as:

$$\mathcal{L}_{\text{class}} \triangleq \sum_{i=1}^{K} (1 - d_i)^2. \tag{4}$$

### 3.2.2 Full Rank Aggregation

To guarantee that all embedding dimensions contribute to class separability, every embedding dimension should play a role in generating the predicted class variables through the aggregate function. It can be achieved by enforcing aggregate function is *full-rank*. The aggregate function satisfies the full-rank property by constraining the weight matrices of the aggregate function to be orthogonal. The *orthogonal regularization* loss is defined as:

$$\mathcal{L}_{\text{ortho}} \triangleq \sum_{l=1}^{L} \|(W_h^{(l)})^T W_h^{(l)} - I\|_F^2, \tag{5}$$

where $W_h^{(l)}$, $I$, and $L$ denote the weight matrix of the aggregate function $h$ at layer $l$, the identity matrix, and the total number of layers in the aggregate function, respectively.

### 3.2.3 Discriminativeness Loss

By combining the class correlation loss and orthogonal regularization loss, as shown on the right side of Figure 2, we ensure that each embedding dimension learns class-discriminative features, meaning that the dimensions acquire the ability to effectively distinguish between different classes, referred to as *discriminativeness*. The overall *discriminativeness loss* is defined as:

$$\mathcal{L}_{\text{disc}} \triangleq \mathcal{L}_{\text{class}} + \lambda_{\text{ortho}} \mathcal{L}_{\text{ortho}}. \tag{6}$$

This combined loss drives the model to maximize class separability while ensuring that all dimensions contribute effectively.

**Overall loss function.** The overall loss, combining the discriminativeness loss to ensure each dimension of the embedding becomes class discriminative with the existing self-supervised dimension contrastive loss, can be expressed as follows:

$$\mathcal{L}_{\text{SupDCL}} \triangleq \mathcal{L}_{\text{disc}} + \mathcal{L}_{\text{DCL}} = \mathcal{L}_{\text{class}} + \lambda_{\text{ortho}} \mathcal{L}_{\text{ortho}} + \lambda_{\text{inv}} \mathcal{L}_{\text{inv}} + \lambda_{\text{decorr}} \mathcal{L}_{\text{decorr}}, \tag{7}$$

where $\lambda_{\text{inv}}$, $\lambda_{\text{decorr}}$, and $\lambda_{\text{ortho}}$ are hyperparameters that balance the respective loss terms.

In our approach, we aim to maximize the mutual information between the class variable $C$ and the learned representation $Y$, building on existing dimension contrastive learning frameworks such as Barlow Twins (Zbontar et al., 2021). This mutual information maximization is achieved through two main objectives: maximizing *diversity* via dimension decorrelation, and maximizing *discriminativeness* via covariate shift-invariant alignment with true class distributions and information-preserving aggregation. The mutual information can be expressed as:

$$I(C;Y) = \underbrace{H(Y)}_{\text{diversity}} - \underbrace{(H(\hat{C} \mid C) + H(Y_{\text{null}}))}_{\text{discriminativeness}}, \tag{8}$$

for theoretical details, refer to Appendix A.

Table 1: **In-domain classification and out-domain transfer learning.** Linear evaluation performance comparison on 10 downstream datasets, for ResNet-50 pre-trained on ImageNet-1K.

| Method | In-domain IN1K | Out-domain | | | | | | | | | | Whole Average |
|---|---|---|---|---|---|---|---|---|---|---|---|---|
| | | CIFAR10 | CIFAR100 | Food | Pets | Flowers | Caltech101 | Cars | Aircraft | DTD | SUN397 | |
| Supervised | 76.1 | 91.7 | 74.4 | 71.2 | 92.3 | 95.4 | 88.7 | 50.0 | 48.6 | 71.9 | 60.4 | 74.6 |
| *Self-supervised Representation Learning:* | | | | | | | | | | | | |
| SimCLR | 69.1 | 90.6 | 71.6 | 68.4 | 83.6 | 91.2 | 90.3 | 50.3 | 50.3 | 74.5 | 58.8 | 72.6 |
| Barlow Twins | 73.2 | 92.9 | 78.3 | 76.1 | 89.9 | 97.7 | 89.9 | 65.4 | 60.2 | 76.9 | 62.9 | 78.5 |
| SwAV | 75.3 | 94.1 | 79.7 | 76.9 | 87.7 | 97.2 | 90.9 | 61.8 | 58.0 | 77.8 | 65.8 | 78.7 |
| MoCo v3 | 71.1 | **94.8** | **80.1** | 73.9 | 90.7 | 96.9 | **91.7** | 65.9 | 61.4 | 75.7 | 63.0 | 78.7 |
| DINO | 75.3 | 93.9 | 79.4 | **78.6** | 89.3 | 97.8 | 90.9 | 67.9 | **62.4** | **77.2** | **65.9** | 79.9 |
| *Supervised Representation Learning:* | | | | | | | | | | | | |
| SupCon | 77.9 | 93.0 | 76.3 | 71.9 | 92.8 | 96.5 | **91.7** | 61.2 | 57.3 | 74.7 | 62.9 | 77.8 |
| PaCo | 78.7 | 91.1 | 70.6 | 64.4 | 92.3 | 88.4 | 87.9 | 37.8 | 34.8 | 68.1 | 58.2 | 70.2 |
| GPaCo | **79.5** | 92.2 | 73.5 | 62.5 | 91.9 | 84.4 | 88.4 | 37.6 | 32.7 | 67.7 | 57.2 | 69.8 |
| SupDCL-1024 (Ours) | 78.2 | 93.8 | 78.5 | 74.3 | **93.1** | 96.6 | **91.7** | 66.5 | 56.2 | 74.6 | 63.9 | 78.9 |
| SupDCL (Ours) | 77.5 | 94.1 | 79.9 | 78.3 | 92.6 | **98.1** | 91.3 | **71.1** | 61.9 | 75.9 | 64.7 | **80.5** |

# 4 EXPERIMENTS

**Baselines.** We compare our method with self-supervised learning methods, including Sim-CLR (Chen et al., 2020), Barlow Twins (Zbontar et al., 2021), SwAV (Caron et al., 2020), MoCov3 (Chen et al., 2021), and DINO (Caron et al., 2021), as well as supervised methods trained with cross-entropy and contrastive learning methods such as SupCon (Khosla et al., 2020), PaCo (Cui et al., 2021), and GPaCo (Cui et al., 2023). We evaluate on both in-domain classification tasks and out-of-domain transfer learning tasks.

**Datasets.** For all tasks, except cross-entropy-based supervised learning, we pre-train a ResNet-50 backbone on ImageNet-1K and evaluate performance using a linear layer. Classification is evaluated using the standard linear protocol on ImageNet-1K, while transfer learning is assessed on 10 downstream tasks (CIFAR10/100 (Krizhevsky et al., 2009), Food (Bossard et al., 2014), Pets (Parkhi et al., 2012), Flowers (Nilsback & Zisserman, 2008), Caltech101 (Fei-Fei et al., 2004), Cars (Krause et al., 2013), Aircraft (Maji et al., 2013), DTD (Cimpoi et al., 2014), SUN397 (Xiao et al., 2010)) with the standard linear transfer protocol (Sun et al., 2017).

**Setup.** In SupDCL, we use a 3-layer non-linear MLP for both the projector and aggregate function, with a 2048-dimensional embedding. The hyperparameters $\lambda_{\text{inv}}$ and $\lambda_{\text{decorr}}$ are set to 1 and 0.0051, respectively, following the settings from Barlow Twins. The orthogonal regularization parameter $\lambda_{\text{ortho}}$ is set to 0.1. For large-scale evaluation, we pre-train a ResNet-50 backbone on the ImageNet dataset using a batch size of 2048 with 4 H100 GPUs. We utilize the LARS optimizer with a weight decay of $1.5 \times 10^{-6}$, training for 1000 epochs with a learning rate of 0.2, which is linearly warmed up over the first 10 epochs, followed by cosine decay. Additionally, we evaluate a SupDCL-1024 variant with a reduced embedding dimension of 1024, a batch size of 1024, and an adjusted $\lambda_{\text{decorr}}$ of 0.04 to optimize decorrelation. For ablation studies, we pre-train a ResNet-18 backbone on CIFAR-100 using a single A6000 GPU. The model is trained with SGD, using a learning rate of 0.03, cosine scheduling with a 10-epoch warm-up, a weight decay of $5 \times 10^{-4}$, and momentum of 0.9. The batch size is set to 256, and the embedding dimension is 128. Details on downstream tasks are provided in Appendix B.

## 4.1 MAIN RESULTS

**In-domain classification & Out-domain transfer learning.** As shown in Table 1, SupDCL demonstrates comparable performance to supervised contrastive learning methods for in-domain classification tasks, while outperforming supervised learning, self-supervised contrastive learning, and self-supervised non-contrastive learning. Additionally, we conduct out-domain transfer learning experiments to evaluate the generalization ability of the SupDCL. SupDCL achieves the highest average performance across all datasets, even though it is trained with labels for fixed tasks. This result highlights the effectiveness of the proposed method in enhancing the discriminativeness of each embedding dimension. Consequently, SupDCL enables more detailed and effective representation learning, leading to superior generalization performance on various datasets.

Table 2: **Decorrelation and discriminativeness of representation dimensions.** Analysis of decorrelated dimensions and discriminative dimensions based on correlations between representation dimensions and class variables. Note that $\tau_r$ is the redundancy threshold, and $\tau_d$ is the discriminativeness threshold. We set $\tau_r = 0.2$ when measuring dimensions that are both decorrelated and discriminative.

| Method | Dim. Corr. ↓ | Decorrelated Dims ↑ | | Discriminative Dims ↑ | | | Decorrelated Discriminative Dims ↑ | | |
|---|---|---|---|---|---|---|---|---|---|
| | ($\times 10^{-3}$) | $\tau_r = 0.1$ | $\tau_r = 0.2$ | $\tau_d = 0.1$ | $\tau_d = 0.2$ | $\tau_d = 0.5$ | $\tau_d = 0.1$ | $\tau_d = 0.2$ | $\tau_d = 0.5$ |
| Supervised | 61.9 | 6 | 39 | 1965 | 197 | 0 | 37 | 4 | 0 |
| Barlow Twins | 31.3 | 48 | **904** | 1320 | 440 | 34 | 628 | 248 | 23 |
| SupCon | 37.5 | 20 | 532 | **2004** | **1298** | 132 | 525 | 325 | 30 |
| SupDCL-1024 | **27.3** | **57** | 699 | 1880 | 1258 | **375** | **641** | **387** | **141** |
| SupDCL | 29.5 | 49 | 645 | 1788 | 950 | 207 | 565 | 309 | 98 |

## 4.2 ANALYSIS

### 4.2.1 REDUNDANCY IN REPRESENTATION

Measuring the number of decorrelated dimensions is crucial as it reflects how effectively a model reduces redundancy across dimensions, ensuring that each dimension contributes unique information rather than duplicating the information captured by other dimensions. For a representation $Y = (R_1, \ldots, R_D)$, we classify two dimensions $R_i$ and $R_j$ as redundant if their correlation $\text{Cor}[R_i, R_j]$ exceeds a redundancy threshold $\tau_r$. Dimensions are considered decorrelated if $\text{Cor}[R_i, R_j] < \tau_r$ for all $i \neq j$. The total number of decorrelated dimensions is calculated by grouping correlated dimensions based on a redundancy threshold $\tau_r$. We construct a graph $G(V, E)$, where $V = \{1, 2, \ldots, D\}$ represents the dimensions of $Y$, and $E = \{(i, j) : |\text{Cor}[R_i, R_j]| \geq \tau_r\}$ defines edges between dimensions $i$ and $j$ if their correlation exceeds $\tau_r$. The total number of decorrelated groups corresponds to the number of connected components in $G$. This ensures that highly correlated dimensions are grouped together, and the resulting count reflects the effective number of independent, decorrelated groups.

As shown in Table 2, with $\tau_r = 0.1$, SupDCL-1024 achieves the highest number of decorrelated dimensions with 57, surpassing Barlow Twins with 48. At $\tau_r = 0.2$, Barlow Twins achieves the highest count with 904, while SupDCL-1024 achieves 699, outperforming SupCon with 532 and the supervised baseline with 39. This demonstrates that SupDCL effectively balances decorrelation while retaining discriminative power, a key advantage over previous methods.

### 4.2.2 DISCRIMINATIVENESS OF REPRESENTATION

To assess discriminativeness, we analyze the correlation between each representation dimension and class variables. We define a dimension as discriminative if it correlates with at least one class variable above a discriminativeness threshold $\tau_d$. Specifically, the number of discriminative dimensions is computed as:

$$\text{\# Discriminative dimensions} = \left| \left\{ i : \max_k |\text{Cor}[R_i, C_k]| > \tau_d \right\} \right|, \tag{9}$$

where $C_k$ represents the class variable for class $k$. As shown in Table 2, at $\tau_d = 0.2$, SupCon has the highest number of discriminative dimensions, with 1298, followed by SupDCL-1024 with 1258, and the supervised model with 197. When the threshold is relaxed to $\tau_d = 0.1$, Barlow Twins achieves the lowest count of 1320. This is expected as Barlow Twins lacks supervision, resulting in fewer dimensions strongly correlated with class variables.

Importantly, when measuring the number of correlated groups that contain at least one discriminative dimension, SupDCL-1024 maintains a strong balance with 641 dimensions at $\tau_d = 0.1$, compared to 525 of SupCon and 628 of Barlow Twins. This highlights the ability of SupDCL to simultaneously reduce redundancy and enhance class discriminability, which is crucial for generalization across tasks.

Table 3: **Effect of discriminativeness.** kNN evaluation Top-1 accuracy (%) of ResNet18 pre-trained on CIFAR-100 with different learning policies.

| Method | In-domain | Out-domain |
|---|---|---|
| Barlow Twins | 52.6 | 39.1 |
| Barlow Twins + Shuffle | 59.2 | 38.0 |
| Barlow Twins + $\mathcal{L}_{CE}$ + $\mathcal{L}_{ortho}$ | 73.8 | 40.8 |
| SupDCL | **74.1** | **42.0** |

Table 4: **Effect of loss function.** kNN evaluation Top-1 accuracy (%) of ResNet18 pre-trained on CIFAR-100 using various loss functions.

| Case | $\mathcal{L}_{cls}$ | $\mathcal{L}_{ortho}$ | $\mathcal{L}_{inv}$ | $\mathcal{L}_{decorr}$ | Accuracy |
|---|---|---|---|---|---|
| (a) | - | - | ✓ | ✓ | 41.3 |
| (b) | ✓ | ✓ | - | ✓ | 72.9 |
| (c) | ✓ | - | ✓ | ✓ | 73.6 |
| Ours | ✓ | ✓ | ✓ | ✓ | **74.2** |

### 4.3 ABLATION STUDY

#### 4.3.1 ALTERNATIVES OF DISCRIMINATIVENESS LOSS

To evaluate the effect of discriminativeness loss, we compare SupDCL with two straightforward supervised extensions of Barlow Twins: (1) Barlow Twins with Cross-Entropy (CE) and a full-rank aggregate function, and (2) Barlow Twins with Class-Wise Shuffle, which replaces positive pairs with data from the same class. Table 3 shows that while both extensions improve performance over the original Barlow Twins, they remain suboptimal compared to SupDCL. Barlow Twins with CE aligns logits with class labels but neglects batch-level relationships, limiting its generalization ability. Barlow Twins with Class-Wise Shuffle learns intra-class consistency but struggles to differentiate between classes effectively. SupDCL, which combines discriminativeness and decorrelation losses, achieves the best performance in both in-domain and out-domain tasks, highlighting its ability to learn diverse and discriminative features.

#### 4.3.2 LOSS FUNCTIONS

We conduct ablation experiments to assess the contribution of each component in the proposed loss function: class correlation loss, orthogonal regularization loss, invariance loss, and decorrelation loss. Table 4 shows that even without the invariance loss, the combination of discriminativeness and decorrelation losses outperforms standard dimension contrastive learning, indicating that discriminativeness loss enables the model to learn class-discriminative features without relying on augmentation invariance. However, omitting orthogonal regularization results in a performance drop, highlighting its importance for maintaining the full-rank property of the aggregate function. SupDCL, with all losses included, achieves the best performance, demonstrating that each component plays a vital role.

#### 4.3.3 AGGREGATE FUNCTION

We analyze the impact of different aggregate function designs on the quality of the learned representations. By varying the number of layers and introducing non-linearity, we assessed which design leads to better performance. As shown in Table 5, deeper, non-linear aggregate functions improved representation quality. However, increasing the number of layers has a greater effect than adding non-linearity, suggesting that sufficient capacity is essential for effectively combining information from the embedding dimensions.

Table 5: **Aggregate function ablations.** kNN evaluation Top-1 accuracy (%) of ResNet18 pre-trained on CIFAR-100 with various aggregate function designs.

| # Layers | Linearity | Accuracy |
|---|---|---|
| 2 | linear | 73.1 |
| 2 | non-linear | 73.5 |
| 3 | linear | 74.1 |
| 3 | non-linear | **74.2** |

#### 4.3.4 ORTHOGONAL REGULARIZATION

We evaluate the effect of orthogonal regularization in Figure 3a. We measure the in-domain classification task performance and the rank of the aggregate function $h$ at the first layer. As $\lambda_{ortho}$ increases from 0 to 0.001, the classification accuracy is improved. The aggregate function satisfies the full-rank property when $\lambda_{ortho}$ is set to 0.1, achieving the highest accuracy. However, increasing $\lambda_{ortho}$ beyond the point where the full-rank property is satisfied may lead to a degradation in perfor-

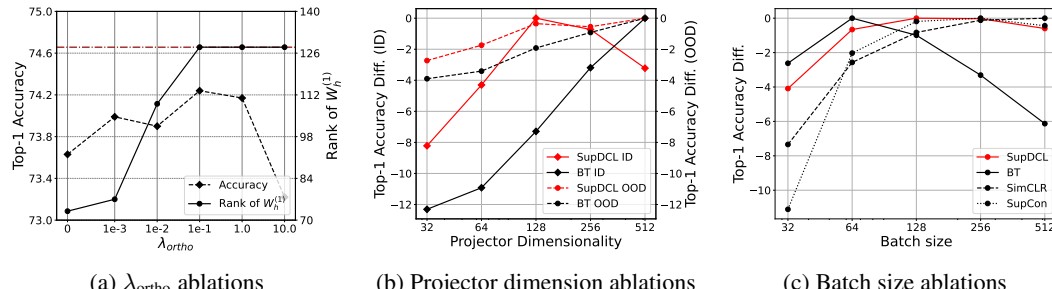

(a) $\lambda_{\text{ortho}}$ ablations  (b) Projector dimension ablations  (c) Batch size ablations

Figure 3: **Ablation studies for SupDCL.** All experiments are conducted using the ResNet-18 encoder on the CIFAR-100 dataset. **(a)**: Top-1 accuracy of SupDCL for in-domain classification task and rank of $W_h^{(1)}$. Note that the maximum value of rank of $W_h^{(1)}$ is 128. **(b)**: Relative Top-1 accuracy of Barlow twins and SupDCL for in-domain classification task and downstream out-domain transfer learning tasks. Each line represents the difference from its respective maximum value. **(c)**: Relative Top-1 accuracy of SimCLR, SupCon, Barlow Twins, and SupDCL for in-domain classification task. Each line represents the difference from its respective maximum value.

mance. These results demonstrate that ensuring the full-rankness of the aggregate function enhances in-domain classification task performance.

### 4.3.5 EMBEDDING DIMENSION

In Figure 3b, we investigate how the dimensionality of the projector affects the performance of in-domain classification and out-domain transfer learning tasks. For out-domain transfer learning tasks, our method achieves the highest accuracy when using the largest output dimension, similar to Barlow Twins. However, our method achieves the best performance with a projection dimension of 128 for in-domain classification task.

### 4.3.6 BATCH SIZE

Figure 3c presents the effect of batch size. SupCon and SimCLR experience significant performance degradation with smaller batch sizes since they are based on contrastive learning. In contrast, dimension contrastive learning methods, Barlow Twins and our method, exhibit strong performance even with smaller batch sizes. Moreover, our method shows the most robust performance across batch size variations.

## 5 DISCUSSION AND CONCLUSION

We introduce Supervised Dimension Contrastive Learning (SupDCL), a novel method that applies supervision to dimension contrastive learning, reducing redundancy across embedding dimensions and improving the discriminativeness of the learned representations. By applying discriminativeness loss, we ensure that each dimension contributes to class discrimination, while simultaneously promoting augmentation invariance and dimensional decorrelation. Our approach demonstrates superior performance across in-domain and out-domain tasks compared to existing self-supervised and supervised methods. Additionally, our analysis confirms that the proposed method functions as intended, further validating its effectiveness.

**Limitations.** Despite these successes, our approach relies on the use of fixed-form class labels for supervision. This limitation suggests potential avenues for future work. Expanding SupDCL to handle more diverse forms of supervision, such as missing or noisy labels, or even non-categorical and multi-task labels, could further enhance its generalizability. Moreover, exploring the integration of continuous or hierarchical labels, and even weak supervision, could broaden the applicability of our method across different domains and tasks. Addressing these challenges will be critical in developing a more robust, flexible, and widely applicable representation learning framework.

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

# A  THEORETICAL JUSTIFICATION

The framework aims to maximize the mutual information $I(C;Y)$ between the class variable $C$ and the learned representation $Y$. This is achieved by:

1. Maximizing the *diversity* via dimension decorrelation.

2. Maximizing the *discriminativeness* via covariate shift-invariant alignment with true class distribution and information-preserving aggregation.

## A.1  MUTUAL INFORMATION DECOMPOSITION: DIVERSITY AND DISCRIMINATIVENESS

The mutual information $I(C;Y)$ between the true class $C$ and the learned representation $Y$ is expressed as:

$$I(C;Y) = H(Y) - H(Y|C), \tag{A1}$$

where $H(Y)$ is the total entropy of $Y$, capturing the diversity of the representation, and $H(Y|C)$ is the conditional entropy of $Y$ given $C$, quantifying the uncertainty in $Y$ given the class label.

We decompose $Y$ using the null-range decomposition into two components: $Y_{\text{range}}$ and $Y_{\text{null}}$. Specifically, $Y_{\text{range}} = AA^{\dagger}Y$ is the projection of $Y$ onto the row space of the aggregation function $A$, where $A^{\dagger}$ is the pseudo-inverse of $A$. $Y_{\text{null}} = (I - AA^{\dagger})Y$ represents the projection of $Y$ into the null space of $A$.

By using this null-range decomposition and the law of total entropy, we can express the conditional entropy $H(Y|C)$ as:

$$H(Y|C) = H(Y_{\text{range}}|C) + H(Y_{\text{null}}|C). \tag{A2}$$

Since $A$ is injective in its range space, there exists a one-to-one mapping between $Y_{\text{range}}$ and the aggregated prediction $\hat{C} = AY_{\text{range}}$, which implies:

$$H(Y_{\text{range}}|C) = H(AY_{\text{range}}|C) = H(\hat{C}|C). \tag{A3}$$

Furthermore, because the aggregation function $A$ is designed to capture all $C$-relevant information in $Y$, the null space contains no information about $C$. Therefore, we treat $H(Y_{\text{null}} \mid C)$ as simply $H(Y_{\text{null}})$, the entropy of the null space projection.

Substituting these relationships into the expression for $I(C;Y)$, we obtain:

$$I(C;Y) = \underbrace{H(Y)}_{\text{diversity}} - \underbrace{(H(\hat{C} \mid C) + H(Y_{\text{null}}))}_{\text{discriminativeness}}, \tag{A4}$$

where:

- $H(Y)$ measures the total information capacity of the learned representation $Y$, which reflects the ability to capture a broad range of features from the input data, as *diversity*.

- $H(\hat{C}|C) + H(Y_{\text{null}})$ quantifies the uncertainty and information loss related to the aggregation process, specifically the model's ability to distinguish between classes and the preservation of class-relevant information, as *discriminativeness*.

## A.2   DIVERSITY $H(Y)$

**Claim:** Correlation between representation dimensions limits the information capacity $H(Y)$ of $Y$.

The entropy of $Y$, assuming a Gaussian distribution, is given by:

$$H(Y) = \frac{1}{2} \log \det(2\pi e \Sigma_Y), \tag{A5}$$

where $\Sigma_Y$ is the covariance matrix of $Y$.

The determinant $\det(\Sigma_Y)$ is the product of its eigenvalues $\{\lambda_1, \lambda_2, \ldots, \lambda_D\}$, so:

$$H(Y) = \frac{1}{2} \sum_{i=1}^{D} \log \lambda_i + \text{constant}. \tag{A6}$$

If the dimensions of $Y$ are highly correlated, some eigenvalues $\lambda_i$ will be close to zero, reducing $H(Y)$ and limiting the effective information capacity of $Y$. By reducing inter-dimension correlation, decorrelation directly enhances the information capacity of $Y$, improving feature diversity.

## A.3   DISCRIMINATIVENESS $H(\hat{C}|C) + H(Y_{\text{NULL}})$

### A.3.1   MINIMIZING $H(\hat{C}|C)$ VIA CORRELATION MAXIMIZATION

**Claim:** Correlation maximization provides covariate shift-invariant distributional alignment, unlike cross-entropy, which relies on pointwise alignment.

The conditional entropy $H(\hat{C}|C)$ is expressed as:

$$H(\hat{C}|C) = \mathbb{E}_x \left[ \mathcal{D}_{\text{KL}}(p(c \mid x) \| q_\theta(c \mid x)) + H(p(c \mid x)) \right], \tag{A7}$$

where $\mathcal{D}_{\text{KL}}(p(c \mid x) \| q_\theta(c \mid x))$ is the Kullback-Leibler (KL) divergence between true class distribution $p(c \mid x)$ and predicted class distribution $q_\theta(c \mid x)$, representing the misalignment between the model's predictions and the true class probabilities, and $H(p(c \mid x))$ is the intrinsic uncertainty of $p(c \mid x)$.

Thus, minimizing $H(\hat{C}|C)$ corresponds to minimizing the KL divergence $\mathcal{D}_{\text{KL}}(p(c \mid x) \| q_\theta(c \mid x))$, which ensures the alignment between the true and predicted class distributions.

However, minimizing the KL divergence directly via *cross-entropy*, which aligns $q_\theta(c|x)$ to $p(c|x)$ pointwise,

$$\mathcal{L}_{\text{CE}} = -\mathbb{E}_{p_{\text{train}}(x)} \left[ \sum_c p(c \mid x) \log \hat{p}(c \mid x) \right], \tag{A8}$$

relies heavily on the specific weighting of $p_{\text{train}}(x)$. In this approach, regions with higher density in $p_{\text{train}}(x)$ dominate the optimization, leading to sensitivity under covariate shift, where $p(x)$ changes but $p(c \mid x)$ remains constant.

In contrast to pointwise alignment via cross-entropy, correlation maximization aligns the global relationship between the true class variable $C$ and the aggregated class variable $\hat{C}$. Correlation maximization aligns the true and predicted class distributions $p(c \mid x)$ and $q_\theta(c \mid x)$ by optimizing the global relationship between the aggregated class variable $\hat{C}$ and the true class variable $C$, with $p(c \mid x)$ and $q_\theta(c \mid x)$ corresponding to the probability distributions of $C$ and $\hat{C}$, respectively.

To achieve this, both $C$ and $\hat{C}$ are normalized to have zero mean and unit variance, thereby reducing the dependency on the distribution $p(x)$. The normalized variables are given by:

$$C' = \frac{C - \mathbb{E}_{p(x)}[C]}{\text{Std}_{p(x)}[C]} \quad \text{and} \quad \hat{C}' = \frac{\hat{C} - \mathbb{E}_{p(x)}[\hat{C}]}{\text{Std}_{p(x)}[\hat{C}]}, \tag{A9}$$

where $\mathbb{E}_{p(x)}[\cdot]$ and $\text{Std}_{p(x)}[\cdot]$ represent the mean and standard deviation of $C$ and $\hat{C}$ under the marginal distribution $p(x)$. This normalization step ensures that the alignment between $C$ and $\hat{C}$ is invariant to the underlying covariate distribution $p(x)$, making it less sensitive to shifts in the input distribution.

The alignment between $C'$ and $\hat{C}'$ is measured using the Pearson correlation coefficient, which quantifies the relationship between the two variables. This is given by:

$$\text{Cor}[C, \hat{C}] = \mathbb{E}_{p_{\text{train}}(x)}[C'\hat{C}']. \tag{A10}$$

Maximizing this correlation encourages a strong alignment between the true and predicted class variables, regardless of the distribution of $x$. Since measuring correlation uses normalization, it mitigates the effects of covariate shift, ensuring a more robust distributional alignment.

### A.3.2 Minimizing $H(Y_{\text{NULL}})$ via Full-rank Aggregation

**Claim**: Full-rank aggregation minimizes information loss by ensuring that $Y_{\text{null}}$ is as low-dimensional as possible, ideally reducing the dimension of the null space.

To minimize information loss in the aggregation process, we aim to reduce the dimensionality of $Y_{\text{null}}$, which represents the part of the learned representation that is not relevant to the class information. The key assumption is that $Y_{\text{null}}$ contains irrelevant or unlearned information, which is commonly modeled as Gaussian noise. Specifically, we assume that $Y_{\text{null}} \sim \mathcal{N}(0, \Sigma_{\text{null}})$, where $\Sigma_{\text{null}}$ is the covariance matrix of the null space. The entropy of $Y_{\text{null}}$ is then expressed as:

$$H(Y_{\text{null}}) = \frac{1}{2} \log\left((2\pi e)^d \det(\Sigma_{\text{null}})\right) \tag{A11}$$

where $d$ is the dimensionality of $Y_{\text{null}}$. This expression captures the information content in the null space. To minimize the information loss, we need to minimize $H(Y_{\text{null}})$, which can be achieved by reducing $d$, the dimensionality of the null space.

The dimensionality $d$ of $Y_{\text{null}}$ depends on the rank of the aggregation function $A : \mathbb{R}^D \to \mathbb{R}^K$, which maps the high-dimensional input representation $Y \in \mathbb{R}^D$ to a lower-dimensional aggregated class variable $\hat{C} \in \mathbb{R}^K$. The rank of the function $A$ determines the amount of class-relevant information retained. If $A$ has rank $r$, the dimensionality of $Y_{\text{null}}$ is $d = D - r$. To minimize $H(Y_{\text{null}})$, we want to reduce $d$, which means maximizing the rank $r$ of $A$. The rank of $A$ is limited by the dimensionality of the output space (which is $K$, the number of classes) and the input space (which is $D$, the dimension of the learned representation). Therefore, the maximum possible rank of $A$ is $r = \min(K, D)$, and in the case of full-rank aggregation, we set $r = K$. This means that the dimensionality of $Y_{\text{null}}$ becomes $d = D - K$.

Thus, by ensuring that the aggregation function $A$ is full-rank, we minimize the dimensionality of $Y_{\text{null}}$, thereby reducing $H(Y_{\text{null}})$ and minimizing the information loss during the aggregation process. This full-rank condition ensures that as much class-relevant information as possible is preserved while irrelevant information is minimized, leading to a more efficient and informative representation.

# B  IMPLEMENTATION DETAILS

## B.1  IN-DOMAIN CLASSIFICATION

For the linear evaluation protocol used after pretraining on ImageNet-1K, a standard linear classifier is trained on top of frozen features. Specifically, the representation is first extracted from the pre-trained model, and then a linear layer is trained on the frozen features to classify ImageNet-1K data. During training, common augmentations such as random resized crops and horizontal flips are applied. The accuracy is reported on the central crop of each validation image. The model is optimized for 100 epochs using stochastic gradient descent (SGD) with a momentum of 0.9, a batch size of 128 and a base learning rate of 0.01.

## B.2  OUT-DOMAIN TRANSFER LEARNING

We perform transfer learning with the linear evaluation on 10 datasets: CIFAR10/100 (Krizhevsky et al., 2009), Food (Bossard et al., 2014), Pets (Parkhi et al., 2012), Flowers (Nilsback & Zisserman, 2008), Caltech101 (Fei-Fei et al., 2004), Cars (Krause et al., 2013), Aircraft (Maji et al., 2013), DTD (Cimpoi et al., 2014), SUN397 (Xiao et al., 2010). We follow the standard linear transfer evaluation protocol (Sun et al., 2017), training linear classifiers on frozen features from $224 \times 224$ resized images without data augmentation. We use L-BFGS to minimize $\ell_2$-regularized cross-entropy loss, selecting the regularization parameter from 45 logarithmically spaced values between $10^{-6}$ and $10^5$ via the validation set. Then, the linear classifier is retrained on both training and validation data with test accuracy.

## C   VISUALIZATION OF REPRESENTATION DIMENSIONS

To validate our analysis of the redundancy and discriminativeness of representation dimensions, as detailed in Table 2, we provide UMAP-based (McInnes et al., 2018) visualizations of the learned representations to clarify the role of each dimension. These visualizations enable low-dimensional projection while preserving the structure of high-dimensional data, offering insight into how the learned representations encode class-relevant information and exhibit decorrelation.

**Dataset and Classes.** We utilize a subset of ImageNet known as the "16-class-ImageNet" (Geirhos et al., 2018), which maps ImageNet categories to 16 entry-level superclasses: airplane, bear, bicycle, bird, boat, bottle, car, cat, chair, clock, dog, elephant, keyboard, knife, oven, and truck. This mapping provides semantically diverse yet interpretable class groups for our analysis.

**Visualization Setup.** Representations are extracted from the validation set of ImageNet-1K and mapped to two dimensions using UMAP with the following hyperparameters: the number of neighbors is set to 5, the minimum distance is 0.8, and the distance metric is Euclidean. For the visualizations, we use thresholds $\tau_d = 0.1$ for discriminative dimensions, and $\tau_r = 0.2$, $\tau_d = 0.1$ for decorrelated discriminative dimensions. The visualizations analyze four categories of dimensions: (1) whole dimensions, (2) discriminative dimensions, (3) non-discriminative dimensions, and (4) decorrelated discriminative dimensions.

For whole dimensions, discriminative dimensions, and decorrelated discriminative dimensions, Figures A1, A2, A3, and A4 demonstrate that the learned representations effectively separate the data based on class information. Among these, the decorrelated discriminative dimensions are sufficient to represent the class-separating information, resulting in similar class separability compared to the whole and discriminative dimensions while avoiding redundancy. In contrast, the non-discriminative dimensions exhibit more diffuse distributions with unclear class boundaries, indicating that these dimensions fail to encode class-relevant information.

We observe that for methods such as Barlow Twins and SupDCL, even the non-discriminative dimensions exhibit a degree of class separability. This can be attributed to their explicit focus on decorrelating dimensions, which allows a larger number of dimensions to encode non-redundant features. This highlights the benefit of decorrelation in promoting feature diversity, even if all dimensions are not fully discriminative.

Finally, we divide the decorrelated discriminative dimensions into four groups and visualize their class separation patterns. Each group exhibits distinct clustering behavior, indicating that these dimensions learn complementary features. This analysis further supports our claim that the number of decorrelated discriminative dimensions serves as an indirect indicator of the diversity in meaningful features.

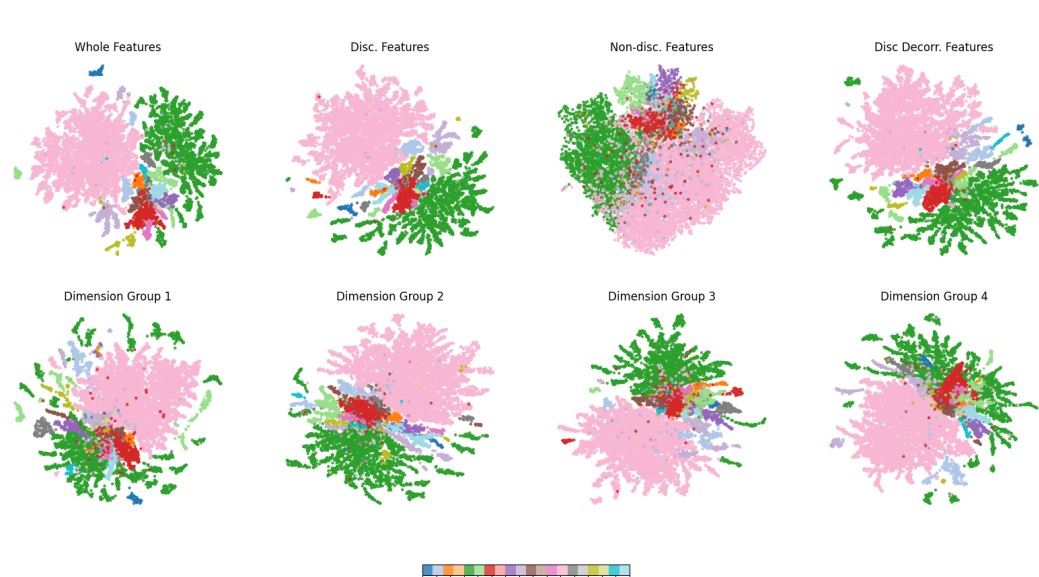

Figure A1: **Visualization of the representations on 16-class-ImageNet learned by SupDCL.** Colors denote superclasses of the dataset. Best viewed in color.

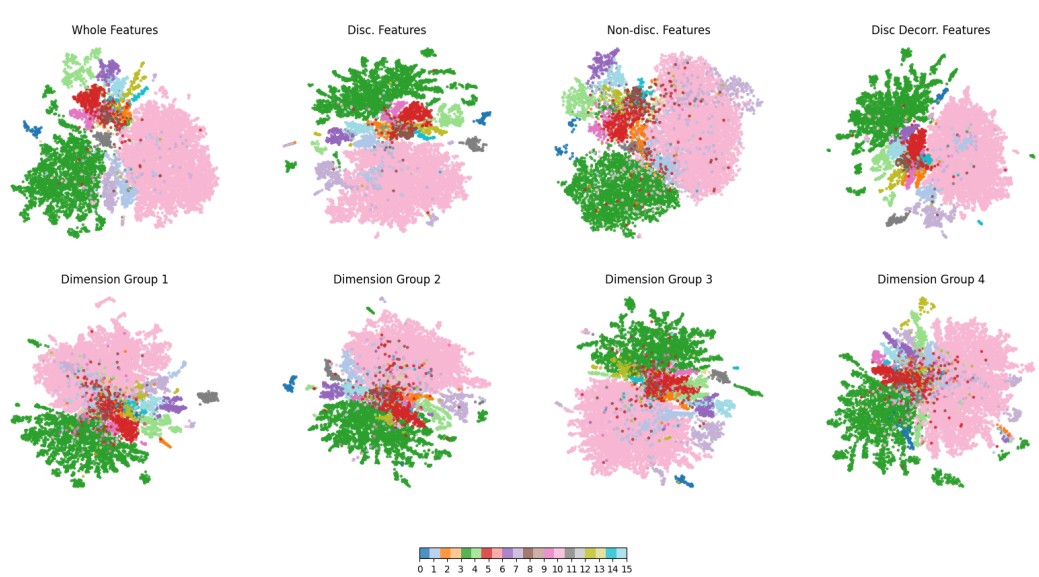

Figure A2: **Visualization of the representations on 16-class-ImageNet learned by Barlow Twins.** Colors denote superclasses of the dataset. Best viewed in color.

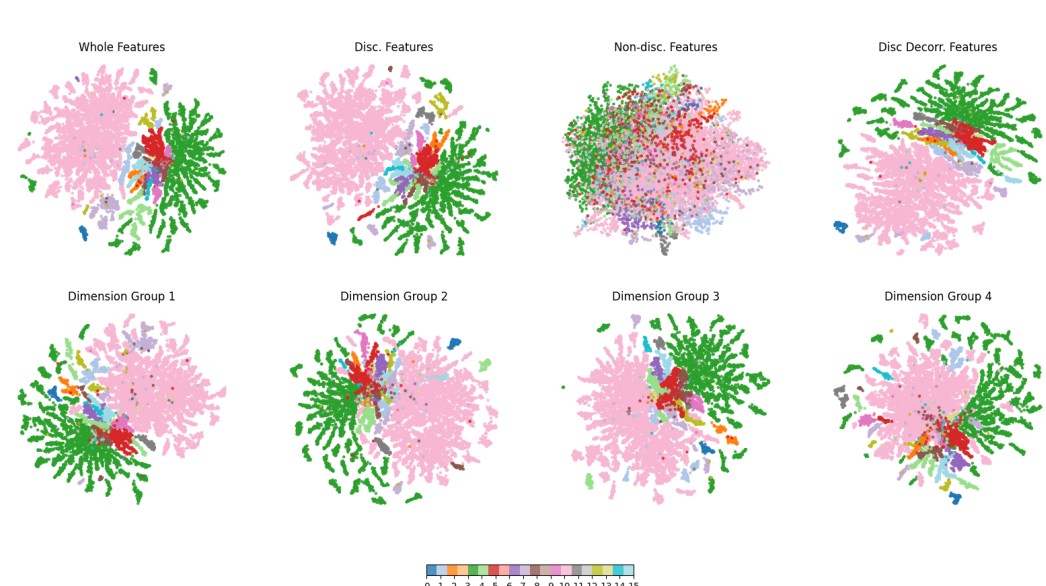

Figure A3: **Visualization of the representations on 16-class-ImageNet learned by SupCon.** Colors denote superclasses of the dataset. Best viewed in color.

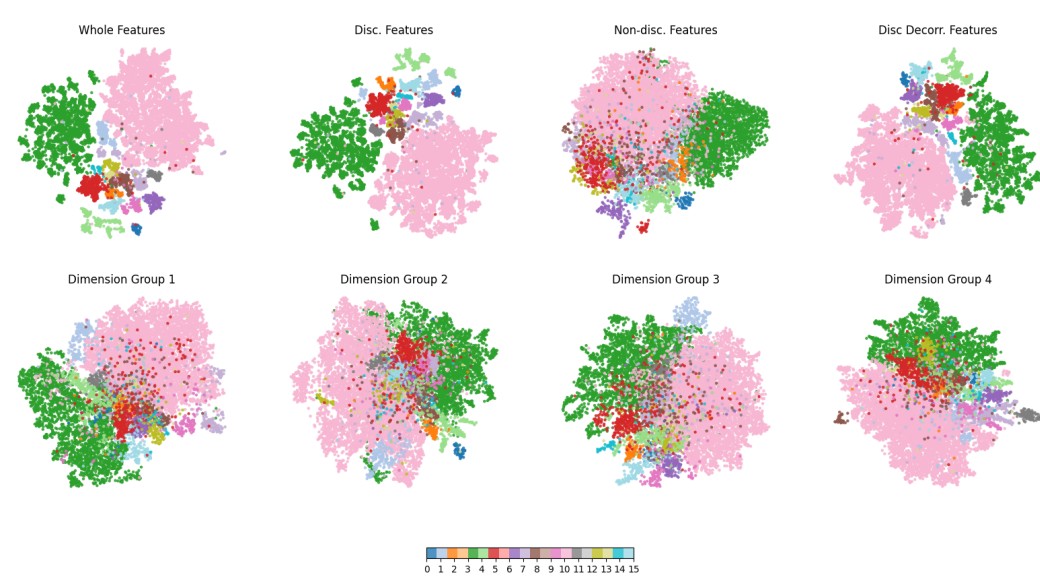

Figure A4: **Visualization of the representations on 16-class-ImageNet learned by Supervised Learning.** Colors denote superclasses of the dataset. Best viewed in color.

