# OpenReview forum: "Supervised Dimension Contrastive Learning"
_ICLR.cc/2025/Conference — ICLR 2025 Conference Withdrawn Submission_

### Official Review · Reviewer_cBsM · 2024-10-31

**Soundness:** 2
**Presentation:** 3
**Contribution:** 2
**Rating:** 5
**Confidence:** 5

**Summary:**

Motivated by Barlow Twins, this paper proposes a method to constrain the feature diversity in supervised contrastive learning. Experimental results show the effectiveness of the proposed method.

**Strengths:**

From experimental section, we can see that the proposed method performance well in OOD scenario. This is interesting and validates to some extent that feature diversity does correlate with the generalizability of feature representations.

**Weaknesses:**

1. The novelty of this paper is really limited. This article does not explain the motivation for CLASS CORRELATION and FULL RANK AGGREGATION to be proposed. At the same time, no intuitive insight or theoretical analysis is given as to why CLASS CORRELATION and FULL RANK AGGREGATION promote supervised comparative learning performance.

2. The related work section is also quite weak. The authors have not done a good job of highlighting the developmental lineage of self-supervised learning, supervised comparative learning, and the connections and differences between the different algorithms. And also did not highlight the advantages and footholds of the method proposed in this paper.

**Questions:**

1. Why can CLASS CORRELATION encourage the embedding dimensions to be not only augmentation-invariant and decorrelated but also discriminative for the supervised classification task? As we can see, CLASS CORRELATION is only a metric of Correlational Relationship.

2. Please give the theoretical analysis or related references to the statement: To guarantee that all embedding dimensions contribute to class separability, every embedding dimension should play a role in generating the predicted class variables through the aggregate function. It can be achieved by enforcing aggregate function is full-rank.

---

> ### Comment · Reviewer_cBsM · 2024-11-23
> **Response to Rebuttal**
>
> **For G2**
>
> 1. As for the sentence: "A is full-rank, the transformation is bijective", it is problematic. Bijection can only be guaranteed if A is A square matrix and A is invertible.
>
> 2. The assumption that labels are published in a Gaussian distribution is somewhat difficult to understand.
>
> 3. The proof of 'Orthogonal Regularization' is built on 'A is square matrix'.
>
> So, I think G2 is unconvincing.
>
>
> **For  CLASS CORRELATION**
>
> What I'm trying to say is that correlation doesn't mean equality. Only the correlation between labels is restricted, and discriminability is not restricted. Therefore, the claim in L229-232 is overclaimed.
>
> **Overall**
>
> Some important concerns have not been addressed well. So, I keep my original rating.

---

> ### Comment · Reviewer_cBsM · 2024-12-03
> **Response to Rebuttal**
>
> Thank you for providing additional proof. However, there are still some important concerns that have not been addressed.
>
> For **Global Response (2-1/4, Revised)**:
>
> 1) Why can Null-Range Decomposition satisfy the law of total entropy?
> 2) Only constrain the **one-to-one mapping** can not lead to the conclusion: $H(Y_{\rm range}|C)=H(AY_{\rm range}|C)=H(\hat C|C)$. This should be bijection.
> 3) $A$ is designed to capture all $C$-relevant information in $Y$. This is a hypothesis, thus, we can use this like a Definitive Conclusion, thus, $H(Y_{\rm null}|C)$ can not been regarded as $H(Y_{\rm null})$.
> 4) Based on 1), 2), and 3), we can not regard $H(\hat C|C) + H(Y_{\rm null})$ as quantifying the uncertainty and information loss related to the aggregation process.
> 5) For Diversity $H(Y)$, the author still assume Y as a Gaussian distribution, this assume is meanful only for Regression Problem, not for Classification Problem.
>
> For **Global Response (2-2/4, Revised)**:
>
> 1) The authors claim that minimizing $H(\hat C|C)$ via Correlation Maximization can be seen as Covariate shift-invariant Distributional Alignment. This is puzzling. The Covariate Shift problem is related to the distribution divergence between the training dataset and the test dataset. However, this paper only focus on training dataset. Also, I read the whole article carefully, and the author do not analyze what confounders caused the distribution shift, nor do he propose a method to extract such confounders. So innovation on this point is overclaimed.
>
> 2) The formulation of $H(Y_{\rm null})$ is still based on the assumption of a Gaussian distribution. Meanwhile, how can you be sure that $H(Y_{\rm null})$ contains information that is not relevant to the task?
>
> For **Contribution**:
>
> 1) The novlty of this paper is incremental. The claim for solving the covariate shift in this paper is problematic.
>
> Thus, I maintain my original rating.

---

### Official Review · Reviewer_UtBt · 2024-11-02

**Soundness:** 2
**Presentation:** 2
**Contribution:** 2
**Rating:** 5
**Confidence:** 4

**Summary:**

This paper enhances Barlow Twins by using the class-correlation loss and orthogonal regularization loss. The class correlation loss uses supervision information, and the orthogonal regularization loss enforces the aggregate function to be full-rank. These additional losses positively affect the overall performance. Through numerical experiments on in-domain classification and out-domain transfer learning tasks, the proposed SupDCL demonstrated superior performance.

**Strengths:**

- The proposed method demonstrated superior performance on in-domain classification and out-domain transfer learning tasks.

**Weaknesses:**

- The class correlation and orthogonal regularization losses have not been investigated in Barlow Twins, but such losses are often investigated in other methods or areas.
- According to Table 4, most of the performance gain of the proposed method compared with Barlow Twins comes from supervision.

**Questions:**

- What is an intuition that the proposed method improved performance on out-domain transfer learning tasks?
- Based on Figure 1, the proposed method inherits the superior out-domain performance from Barlow Twins. The other regularization or loss terms are likely to obtain improvement similar to the proposed method from Barlow twins.
- Are there any theoretical insights about the performance of the proposed method?
- Can we use the proposed method without the class correlation loss as a self-supervised learning method? If so, does this outperform the existing methods?
- Regarding Table 4, is there any discussion of the performance of Barlow Twins plus the orthogonal regularization loss?
- In Table 3(b), the existing methods increase performance as the projector dimensionality increases, but the proposed method decreases after 128. Is there any interpretation for that?

---

### Official Review · Reviewer_T162 · 2024-11-04

**Soundness:** 2
**Presentation:** 3
**Contribution:** 2
**Rating:** 5
**Confidence:** 5

**Summary:**

The paper introduces Supervised Dimension Contrastive Learning (SupDCL), which combines supervision with dimension-wise contrastive learning. It targets limitations in supervised contrastive learning (high cross-correlation and limited feature diversity) by reducing redundancy in embedding dimensions while enhancing class discriminability.

**Strengths:**

- The integration of dimension contrastive learning with supervision to enhance feature diversity makes sense. SupDCL’s approach to reducing cross-correlation among dimensions to improve generalization, particularly in transfer learning tasks, is a well-justified.
- The evaluation covers both in-domain and out-domain tasks on different datasets, providing a clear comparison of SupDCL against multiple baselines, including self-supervised and supervised learning methods.
- Extensive ablation studies that analyze each component of SupDCL’s loss function.
- The paper acknowledges the limitations of SupDCL, specifically its reliance on fixed-form class labels. It provides suggestions for handling these limitations through future work.

**Weaknesses:**

- The paper introduces orthogonal regularization and decorrelation strategies to improve feature diversity, but it lacks a theoretical foundation to show why these particular choices are necessary. While the empirical results show improved performance, a lack of formal justification leaves the effectiveness of these strategies unclear.
- While SupDCL outperforms SupCon in out-doman tasks, it underperforms SupCon in in-doman tasks.
- SupDCL borrows heavily from existing redundancy reduction techniques (Barlow Twins) without significantly advancing the state of contrastive learning. The contribution of introducing supervision to dimension contrastive learning is incremental and could be viewed as a straightforward application of existing techniques.

**Questions:**

- Why is feature discriminative inportant for supervised contrastive learning? Figure 1 and experiments show that SupDCL underperforms SupCon in in-doman tasks.
- What is $L_{inv}$ and $L_{decorr}$ is Eq (7)? They are not explained.

---

### Official Review · Reviewer_cMuw · 2024-11-05

**Soundness:** 3
**Presentation:** 3
**Contribution:** 3
**Rating:** 5
**Confidence:** 5

**Summary:**

This paper argues that representations learned by supervised contrastive learning methods have limited feature diversity and high cross-correlation between dimensions. The authors propose an approach to learn representations that are both augmentation-invariant and dimension-wise decorrelated, while also being discriminative for classification. They achieve this by extending Barlow Twins, a dimension contrastive learning model, to supervised dimension contrastive learning. Specifically, they introduce a discriminativeness loss instead of the cross-entropy loss, consisting of a class correlation loss and an orthogonal regularization loss. Experimental results show that the proposed SupDCL performs well on both in-domain supervised classification tasks and out-of-domain transfer learning tasks.

**Strengths:**

The idea of this paper is to combine SupCon (supervised contrastive learning) with Barlow Twins (decorrelated features). The specific approach involves introducing a discriminativeness loss to leverage the supervised labels. The motivation and approach are presented very clearly. The authors also provide experiments, analysis, and evaluation metrics to demonstrate the effectiveness of the proposed method.

**Weaknesses:**

I believe the experiments could be strengthened, and there are some minor points that would enhance the clarity of the paper. These are outlined in more detail in the Questions section.

**Questions:**

1. This work aims to extend Barlow Twins to a supervised version. A straightforward extension model could be a combination of Barlow Twins and SupCon, where, instead of generating two augmentations $X^A$ and $X^B$ from image $X$, $X^B$ is selected from another image that shares the same class as image $X$. In Table 2, I would like to see the results of this model.

2. Another straightforward extension is to use a cross-entropy loss function instead of the proposed discriminativeness loss function to leverage the supervised labels. While you have compared this model in Table 3, comparing the results in Table 1 and Table 2 would also be worthwhile.

3. In Table 3, you mention the representation space and embedding space; however, in the methods section and Figure 2, there is no reference to the representation space. You have defined the embedding space, $Z$, as an $M-dimensional$ vector, but it is unclear which vector you refer to as the representation vector.

4. Similarly, Section 4.2 does not clarify what the representation $Y$, a $D-dimensional$ vector, refers to.

5. In lines 371-374, when $t_d = 0.1$, Barlow Twins achieves 1320, which seems to be the lowest count rather than the highest as described. Is there a mistake here?

6. Visualization: Could you visualize the embedding space or representation space (with clearer references if possible) of different methods, along with the different dimensions of those representations? This would help to better understand how the proposed method achieves improved decorrelation and discriminative representations.


Minors:
1. It would be better to mark all the best numbers in all tables in bold.

2. In Table 2, add arrows to indicate whether larger or smaller values are better, and mark the best value in bold.

3. Figure 1 can be split into two separate figures, as the third sub-figure is not on the same scale as the other two.

4. In lines 353-358, 645 (SupDCL) vs. 532 (SupCon) is described as outperforming, while 645 (SupDCL) vs. 904 (Barlow Twins) is described as comparable?

---

### Author Response · Authors · 2024-11-28
**Clarification of Our Framework**

Dear Reviewers,

We thank the reviewers for their valuable feedback, which guided us to strengthen the **theoretical and empirical justification** of our proposed dimension-based framework. Initially, we focused on the motivations behind our approach. Now, we theoretically and empirically demonstrate that all three components addresses specific challenges. Specifically, we decompose the mutual information between class labels and learned representations into two components (refer to Revised G2): **diversity** and **discriminativeness**, and together, they form a **comprehensive framework** for robust generalization. All justifications have been revised and reflected in the updated version of the paper.

$I(C;Y)$

$=\underbrace{H(Y)}_{\text{diversity}}$ “the total information capacity of the learned representation $Y$”

$-\underbrace{(H(\hat{C}|C)+H(Y_\text{null}))}_{\text{discriminativeness}}$ “the uncertainty and information loss related to the aggregation process”


1. Dimension Decorrelation - Maximizing $H(Y)$:
    - Reduces inter-dimension correlation using Barlow Twins (BT), theoretically shown to **increase information capacity of representation** (Revised G2 of the global response and Appendix A.2).
    - Empirically, it improves generalization performance, as evidenced by the relationship between dimension correlation and whole-domain accuracy (Figure 1) and the effectiveness of BT+CE over CE alone (Table B).


2. Class Correlation - Minimizing $H(\hat{C}|C)$:
    - A novel method for robust distributional alignment, theoretically shown to **be invariant to covariate shift**, and minimize distance between predicted and true class distributions (Revised G2 of the global response and Appendix A.3.1).
    - Empirically, class correlation improves in-domain and out-domain performance, consistently outperforming CE across all settings (Table B).


3. Full-rank Aggregation - Minimizing $H(Y_\text{null})$:
    - Preserves class-relevant information of representation, theoretically shown to **minimize information capacity of null space** (Revised G2 of the global response and Appendix A.3.2).
    - Empirically, comparisons such as BT+CE vs. BT+CE+Orthogonal and SupDCL w/o Orthogonal vs. SupDCL confirm its significant contribution to both in-domain and out-domain performance (Table B).

Combining these components, our method achieves **state-of-the-art whole-domain generalization**, outperforming all supervised and self-supervised baselines across in-domain and out-domain tasks (Table C).


**Table C**. In-domain classification and out-domain transfer learning. Linear evaluation performance comparison on 10 downstream datasets, for ResNet-50 pre-trained on ImageNet-1K.

| **Method** | **In-domain (IN1K)** | **Average out-domain (10 datasets)** | **Whole-domain (all 11 datasets weighted equally)** | **Whole-domain (50% in-domain, 50% averaged out-domain)** |
| --- | --- | --- | --- | --- |
| Supervised | 76.1 | 74.5 | 74.6 | 75.3 |
| *Self-supervised Representation Learning:* |  |  |  |  |
| SimCLR | 69.1 | 73.0 | 72.6 | 71.0 |
| Barlow Twins | 73.2 | 79.0 | 78.5 | 76.1 |
| SwAV | 75.3 | 79.0 | 78.7 | 77.2 |
| MoCo v3 | 71.1 | 79.4 | 78.7 | 75.3 |
| DINO | 75.3 | 80.3 | 79.9 | 77.8 |
| *Supervised Representation Learning:* |  |  |  |  |
| SupCon | 77.9 | 77.8 | 77.8 | 77.9 |
| PaCo | 78.7 | 69.4 | 70.2 | 74.1 |
| GPaCo | **79.5** | 68.8 | 69.8 | 74.2 |
| **SupDCL-1024 (Ours)** | 78.2 | 78.9 | 78.9 | 78.6 |
| **SupDCL (Ours)** | 77.5 | **80.8** | **80.5** | **79.2** |


**Table B.** Comparison of SupDCL and straightforward extensions of Barlow Twins with cross-entropy loss

| **Method** | **In-domain (CIFAR-100)** | **Average Out-domain** |
| --- | --- | --- |
| Barlow Twins | 52.6% | 39.1% |
| Barlow Twins + CE | 72.6% | 39.6% |
| Barlow Twins + CE with Orthogonalization | 73.8% | 40.8% |
| SupDCL w/o Orthogonalization | 73.8% | 41.0% |
| SupDCL (Ours) | **74.2%** | **42.0%** |

---

### Note · Authors · 2025-01-25

I have read and agree with the venue's withdrawal policy on behalf of myself and my co-authors.